# Random Regression Analysis of Calving Interval of Japanese Black Cows

**DOI:** 10.3390/ani11010202

**Published:** 2021-01-15

**Authors:** Shinichiro Ogawa, Masahiro Satoh

**Affiliations:** Graduate School of Agricultural Science, Tohoku University, Sendai 980-8572, Miyagi, Japan; masahiro.satoh.d5@tohoku.ac.jp

**Keywords:** breeding value, calving interval, genetic correlation between ages, heritability, Japanese black cattle, random regression model, repeatability model

## Abstract

**Simple Summary:**

Genetic parameters for the calving interval of Japanese Black cows were estimated by using a random regression model and a repeatability model. Legendre polynomials based on age at previous calving, ranging from 18 to 120 months, were used as sub-models for random regression analysis. The estimated heritability for the calving interval was low and was similar between the models. The estimated genetic correlation between ages was always higher than >0.8. Spearman’s rank correlation of the estimated breeding values between the two models was ≥0.97 for cows with their own records and ≥0.94 for sires of these cows. Therefore, this study supports the validity of fitting a repeatability model to the records of the calving interval of Japanese Black cows for breeding value evaluation. Our results could contribute to determining strategies for selection and management of Japanese Black cattle.

**Abstract:**

We estimated genetic parameters for the calving interval of Japanese Black cows using a random regression model and a repeatability model. We analyzed 92,019 calving interval records of 36,178 cows. Pedigree data covered 390,263 individuals. Age of cow at previous calving for each record ranged from 18 to 120 months. We used up to the second-order Legendre polynomials based on age at previous calving as sub-models for random regression analysis, and assumed a constant error variance across ages. Estimated heritability was 0.12 to 0.20 with the random regression model and 0.17 with the repeatability model. With the random regression model, the estimated genetic correlation between ages was ≥0.87, and those between 24 and 36 months, 24 and 84 months, and 36 and 84 months were 0.99, 0.95, and 0.97, respectively. Spearman’s rank correlation between breeding values of 36,178 cows with their own records estimated by the random regression model with those estimated using the repeatability model was ≥0.97, and the rank correlation was ≥0.94 for 314 sires of these cows. These results support the validity of fitting a repeatability model to the records of the calving interval of Japanese Black cows for evaluation of breeding values.

## 1. Introduction

Japanese Black beef cattle are well known to excel in meat quality including the degree of marbling [1,2]. Wagyu cattle consist of four Japanese native breeds: Japanese Black cattle (the primary breed), Japanese Brown cattle, Japanese Shorthorn cattle, and Japanese Polled cattle. In Japan, elite sires are selected for artificial insemination in each region or prefecture; since 1991, when beef importation was liberalized, breeding values for major carcass traits have been routinely evaluated by the Best Linear Unbiased Prediction (BLUP) method [3,4] using the carcass performance of progenies and deep pedigree information [5,6]. To accomplish stable and efficient beef production, it is important to improve female reproductive efficiency in Japanese Black cattle [7,8]. The calving interval (CI) is a female reproductive efficiency trait that can be relatively easily recorded. A shortened CI is directly associated with an increased number of progenies, so the economic importance of the CI in Japanese Black beef production is considered to be substantial [9]. Using a repeatability model, a low heritability has been repeatedly estimated for the CI in Japanese Black cow populations of different prefectures [7].

When using a repeatability model, it is assumed that the additive genetic variance is constant and the genetic correlation is 1 over time [10]. If an inappropriate model is used for evaluation of breeding value, the genetic gain could be lower than expected [11]. CI can be analyzed as a longitudinal trait or “function valued-trait” using a random regression model [12,13,14]. Gutierrez (2010) [15] analyzed 1852 records for days open (DO) for 766 Holstein dairy cows from 20 to 90 months of age using pedigree information on 1644 individuals and a random regression model that used up to first-order Legendre polynomials based on calving age as sub-models, and reported that the genetic correlation between ages was 0.99. On the other hand, Panetto et al. (2012) [16] estimated lower genetic correlations from random regression analysis of 4409 CI records obtained from 1114 Guzerat dairy cows from 26 to 150 months of age with up to third-order Legendre polynomials based on calving age. The validity of fitting a repeatability model to CI records of Japanese Black cows should be investigated also. Using random regression models, Aziz et al. (2005) analyzed body weight of Japanese Black calves from birth to 356 days of age [17], and Nishida et al. (2006) analyzed the number of services per conception of Japanese Black cows [18]. However, no study has reported the results of random regression analysis for the CI in Japanese Black cows. Therefore, aiming to assess the validity of fitting a repeatability model, we analyzed CI records of Japanese Black cows using a random regression model.

## 2. Materials and Methods

### 2.1. Ethics Statement

Approval of the Animal Care and Use Committee was not required for this study because the data were acquired from an existing database.

### 2.2. Phenotype and Pedigree Data

The initial data set comprised market records of 295,536 Japanese Black calves shipped to the calf market in Miyagi prefecture, one of the northeast prefectures of Japan, from 2001 to 2016. These calves were derived from either artificial insemination or embryo transfer. The initial data set included dates at service and birth, the identification number of each calf’s dam and up to three generations of ancestors. We constructed the pedigree information of 390,263 individuals form the initial data set and calculated the inbreeding coefficient of each individual using the CoeFR program [19].

All calves overlapping in gestation period were removed from the initial data set owing to the possibility of embryo transfer. Using the records of the dates at service and birth of the remaining calves, the phenotypic records for gestation length (GL) and DO were calculated. Phenotypic CI records (days) were the sum of DO and the following GL. Values of the CI with DO < 21 or >364 days and those with GL < 260 or >310 days were removed [20,21]. Only farms from which more than one calf were reared and shipped were included. In all, 92,019 CI records of 36,178 cows were used. The number of sires of these 36,178 cows was 314. Age of dam at previous parity for each record ranged from 18 to 120 months. It should be noted that the information about the parity of the cows was not available in this study.

### 2.3. Statistical Analysis

The following single-trait repeatability animal model was used:y=Xb+Tf+Za+Wpe+e
where **y** is the vector of phenotypic CI records; **b** is the vector of macro-environment effects of year at previous calving (14 levels), month at previous calving (12 levels), sex of calf (2 levels), inbreeding coefficient of cow (linear covariate), and age at previous calving (linear and quadratic covariates); **f** is the vector of the effects of farm (202 levels); **a** is the vector of breeding values of cows; **pe** is the vector of permanent environmental effects of cows; **e** is the vector of random errors; and **X**, **T**, **Z**, and **W** are the design matrices relating vectors **b**, **f**, **a**, and **pe**, respectively, to **y** [20]. We included farm effect f as random effects, according to previous studies for Japanese Black cattle [20,22], because the sizes of some farms were small in this study. The vectors **f**, **a**, **pe**, and **e** were assumed to follow a multivariate normal distribution with mean and variance–covariance structures of: $$E\left[\begin{array}{c} f\\ a\\ pe\\ e \end{array}\right]=\left[\begin{array}{c} 0\\ 0\\ 0\\ 0 \end{array}\right]\;\mathrm{and}\;\mathrm{var}\left[\begin{array}{c} f\\ a\\ pe\\ e \end{array}\right]=\left[\begin{array}{cccc} I\sigma_{f}^{2} & 0 & 0 & 0\\ 0 & A\sigma_{a}^{2} & 0 & 0\\ 0 & 0 & I\sigma_{pe}^{2} & 0\\ 0 & 0 & 0 & I\sigma_{e}^{2} \end{array}\right]$$
where $\sigma_{f}^{2}$ is the variance of farm effect; $\sigma_{a}^{2}$ is the additive genetic variance; $\sigma_{pe}^{2}$ is the permanent environmental variance; $\sigma_{e}^{2}$ is the error variance; **A** is the additive relationship matrix constructed from pedigree information of 390,263 individuals; and **I** is the identity matrix.

We also analyzed the CI records using the following random regression model: $$y_{ijk}=F_{ijk}+\sum_{l=0}^{2}\varphi_{l}\left(j\right)b_{l}+\sum_{l=0}^{2}\varphi_{l}\left(j\right)f_{kl}+\sum_{l=0}^{2}\varphi_{l}\left(j\right)a_{il}+\sum_{l=0}^{2}\varphi_{l}\left(j\right)pe_{il}+e_{ijk}$$
where *y_ijk_* is the CI of cow *i* on farm *k* at *j* months old at previous calving; *F_ijk_* is the sum of the macro-environment effects (year at a previous calving, month at a previous calving, sex of calf, and inbreeding coefficient of dam (linear)); *b_l_* is the fixed regression coefficient of the *l*th-order Legendre polynomial at *j* months old at previous calving $\varphi_{l}\left(j\right)$ (*l* = 0, 1, or 2) considering the effect of age at previous calving; *f_kl_* is the random regression coefficient of $\varphi_{l}\left(j\right)$ for the effect of farm *k*; *a_il_* is the random regression coefficient of $\varphi_{l}\left(j\right)$ for breeding value of cow *i*; *pe_il_* is the random regression coefficient of $\varphi_{l}\left(j\right)$ for the permanent environmental effect of cow *i*; and *e_ijk_* is the error. This model can also be described in matrix notation as:$$y=Xb+\sum_{l=0}^{2}T_{l}f_{l}+\sum_{k=0}^{2}Z_{l}a_{l}+\sum_{l=0}^{2}W_{l}pe_{l}+e.$$

The vectors **f***_l_*, **a***_l_*, and **pe***_l_* were assumed to follow a multivariate normal distribution with mean of 0 and variance–covariance structure of, respectively:$$\mathrm{var}\left[\begin{array}{c} f_{0}\\ f_{1}\\ f_{2} \end{array}\right]=\left[\begin{array}{ccc} \sigma_{f0}^{2} & \sigma_{f01} & \sigma_{f02}\\ \sigma_{f01} & \sigma_{f1}^{2} & \sigma_{f12}\\ \sigma_{f02} & \sigma_{f12} & \sigma_{f2}^{2} \end{array}\right]⊗I=K_{f}⊗I,$$
$$\mathrm{var}\left[\begin{array}{c} a_{0}\\ a_{1}\\ a_{2} \end{array}\right]=\left[\begin{array}{ccc} \sigma_{a0}^{2} & \sigma_{a01} & \sigma_{a02}\\ \sigma_{a01} & \sigma_{a1}^{2} & \sigma_{a12}\\ \sigma_{a02} & \sigma_{a12} & \sigma_{a2}^{2} \end{array}\right]⊗A=K_{a}⊗A,$$
$$\mathrm{var}\left[\begin{array}{c} pe_{0}\\ pe_{1}\\ pe_{2} \end{array}\right]=\left[\begin{array}{ccc} \sigma_{pe0}^{2} & \sigma_{pe01} & \sigma_{pe02}\\ \sigma_{pe01} & \sigma_{pe1}^{2} & \sigma_{pe12}\\ \sigma_{pe02} & \sigma_{pe12} & \sigma_{pe2}^{2} \end{array}\right]⊗I=K_{pe}⊗I,$$
and the error variance $\sigma_{e}^{2}$ was assumed to be constant over age. When $\Phi (j)={\left[\begin{array}{ccc} \varphi_{0}\left(j\right) & \varphi_{1}\left(j\right) & \varphi_{2}\left(j\right) \end{array}\right]}^{\prime }$, the estimated heritability and repeatability at *j* months old on previous calving, respectively, can be calculated as follows:$$\frac{\Phi {\left(j\right)}^{\prime }{\hat{K}}_{a}\Phi \left(j\right)}{\Phi {\left(j\right)}^{\prime }{\hat{K}}_{f}\Phi \left(j\right)+\Phi {\left(j\right)}^{\prime }{\hat{K}}_{a}\Phi \left(j\right)+\Phi {\left(j\right)}^{\prime }{\hat{K}}_{pe}\Phi \left(j\right)+{\hat{\sigma }}_{e}^{2}}\;\mathrm{and}$$
$$\frac{\Phi {\left(j\right)}^{\prime }{\hat{K}}_{a}\Phi \left(j\right)+\Phi {\left(j\right)}^{\prime }{\hat{K}}_{pe}\Phi \left(j\right)}{\Phi {\left(j\right)}^{\prime }{\hat{K}}_{f}\Phi \left(j\right)+\Phi {\left(j\right)}^{\prime }{\hat{K}}_{a}\Phi \left(j\right)+\Phi {\left(j\right)}^{\prime }{\hat{K}}_{pe}\Phi \left(j\right)+{\hat{\sigma }}_{e}^{2}}.$$

The estimated value of the additive genetic correlation between *i* and *j* months of age at previous calving is:$$\frac{\Phi {\left(i\right)}^{\prime }{\hat{K}}_{a}\Phi \left(j\right)}{\sqrt{\Phi {\left(i\right)}^{\prime }{\hat{K}}_{a}\Phi \left(i\right)}\sqrt{\Phi {\left(j\right)}^{\prime }{\hat{K}}_{a}\Phi \left(j\right)}}.$$

Variance components were estimated via Gibbs sampler using the gibbs3f90 program [23]. A total chain length of 110,000 rounds was run in a single long chain. After the first 10,000 samples were discarded as the burn-in, 1 in every 20 samples was stored. We assessed the Gibbs sampling chains by visual inspection. Parameter estimates and their standard errors (SEs) were obtained by calculating the averages and standard deviations (SDs) of the 5000 samples stored. Eigenvalue decomposition of the estimated variance–covariance component matrices for random regression coefficients, namely ${\hat{K}}_{f}$, ${\hat{K}}_{a}$, and ${\hat{K}}_{pe}$, were carried out using the eigen function in R software [24].

We calculated Spearman’s rank correlation between estimated breeding values (EBVs) of 36,178 cows with their own CI records by the repeatability model and those estimated by the random regression model. The rank correlation of 314 sires of the 36,178 cows was also calculated. The accuracy of the EBV of individual *i* was calculated as:$$\sqrt{1-\frac{PEV_{i}}{\left(1+F_{i}\right){\hat{\sigma }}_{a}^{2}}},$$
where *PEV_i_* is the prediction error variance (PEV) of the EBV of individual *i*, and *F_i_* is the inbreeding coefficient of individual *i*. The squared posterior SD of the EBV was used as the PEV. The accuracy of the EBV of cow *i* was compared with the theoretical selection accuracy of a cow for the CI calculated as:$$\sqrt{\frac{n{\hat{h}}^{2}}{1+\left(n-1\right){\hat{c}}^{2}}},$$
where *n* is the number of CI records for cow *i*; and ${\hat{h}}^{2}$ and ${\hat{c}}^{2}$ are the heritability and repeatability of the CI, respectively, estimated by the repeatability model. Assuming that the mean of age at first calving was 24 months and calving interval was 12 months, the expected number of CI records for a cow at *t* months old can be expressed as (*t* ≥ 24):$$\left[\frac{t-24}{12}\right],$$
where [ ] is the Gauss symbol. Therefore, the theoretical selection accuracy of a cow at *t* months old for the CI can be expressed as:$$\sqrt{\frac{\left[\frac{t-24}{12}\right]{\hat{h}}^{2}}{1+\left(\left[\frac{t-24}{12}\right]-1\right){\hat{c}}^{2}}}.$$

Assuming that all progenies of a cow are fattened and slaughtered at 30 months, the expected number of slaughtered progenies of a cow at *t* months old can be expressed as:$$\left[\frac{t-42}{12}\right].$$

Therefore, the theoretical selection accuracy of a cow at *t* months old for carcass trait with heritability of ${\hat{h}}^{2}$ by progeny testing can be calculated as:$$\sqrt{\frac{\left[\frac{t-42}{12}\right]{\hat{h}}^{2}}{4+\left(\left[\frac{t-42}{12}\right]-1\right){\hat{h}}^{2}}}.$$

We set the heritability of carcass trait to 0.4, 0.5, and 0.6 [7].

## 3. Results and Discussion

### 3.1. Basic Statistics

The distribution of CI records was positively skewed (Figure 1a). We and previous studies for Japanese Black cows analyzed the CI records without data transformation [20,22,25,26,27,28,29,30,31]. Data transformation might bring more desirable results, although re-transformation of the results would be needed. Therefore, in the future, a study exploring the way to properly transform the CI records of Japanese Black cows may be required. The CI records had a mean of 402.1 days, SD of 66.7 days, a minimum value of 295 days, and a maximum value of 661 days. Previous studies of Japanese Black cattle showed average values of CI records to be from 390 to 400 and a SD similar to ours [20,22,25,26,27,28,29,30,31]. The mode of CI records was 365 days, 37 days shorter than the mean, and 36.0% of total CI records were shorter than the mode. The number of CI records per age at previous calving had a multimodal distribution (Figure 1b). Each peak was located around multiples of 12 months (24, 36, 48, …; Figure 1b), which corresponds approximately to dam parity.

### 3.2. Variance Components, Heritability, and Repeatability

The cumulative contribution ratio of principal components has been used for determining the highest order of Legendre polynomials [32,33]; the number at which it first exceeds 98% seems to be reasonable [33,34]. For ${\hat{K}}_{f}$, the first, second, and third principal components contributed 83.8%, 11.9%, and 4.4%, respectively. For ${\hat{K}}_{a}$, they contributed 97.5%, 1.5%, and 1.0%, respectively. For ${\hat{K}}_{pe}$, they contributed 65.9%, 24.4%, and 9.8%, respectively. The cumulative contribution ratio of the first and second principal components exceeded 90% for ${\hat{K}}_{f}$ and ${\hat{K}}_{pe}$ and reached 99.0% for ${\hat{K}}_{a}$. Therefore, it is likely that there is no urgent need to use the third- or higher-order Legendre polynomials for modelling breeding values in this study.

With the random regression model, the estimated value ± SE ranged from 589.7 ± 78.2 to 927.7 ± 90.5 for additive genetic variance, 273.0 ± 70.5 to 957.6 ± 146.5 for the permanent environmental variance, 9.8 ± 3.8 to 24.2 ± 13.3 for the variance of farm effect, and 3375.9 ± 29.0 for the error variance (Figure 2, Appendix A). With the repeatability model, the estimated value ± SE was 810.1 ± 59.8 for additive genetic variance, 276.5 ± 41.3 for the permanent environmental variance, 10.2 ± 3.0 for the variance of farm effect, and 3535.9 ± 20.9 for the error variance. The SEs were larger for random regression analysis, mainly owing to model complexity. The estimated value of additive genetic variance decreased gradually when the age at previous calving approached its minimum (18 months) and maximum (120 months), and that of permanent environmental variance increased sharply when the age approached its minimum and maximum (Figure 2a,b). It has been reported that the estimated variance components might be biased (artifact) at intervals with fewer phenotypic records when using Legendre polynomials as sub-models [35,36,37]; this problem could also be relevant to this study because the numbers of records at 18 and 120 months were very few (Figure 1b). Except around the minimum and maximum values of age at previous parity, the value of additive genetic variance estimated by random regression analysis was similar to that estimated by repeatability model analysis (Figure 2a). These results imply that the heterogeneity of additive genetic variance seems to be low for the CI of Japanese Black cows.

The repeatability model estimated heritability as 0.17, repeatability as 0.23, and the ratio of variance of farm effect to phenotypic variance as <0.01; heritability and repeatability were slightly higher and the ratio of variance was lower than those in previous studies of the CI of Japanese Black cows in different subpopulations [20,22,27,31]. Possible reasons for the differences among studies are differences in genetic background among subpopulations and in the method of data editing. The unavailability of information about body condition score and movement of cows between farms could also affect the results [20]. Estimated low proportion of farm effect variance might imply the possibility of removing the farm effects from the model, although a thoughtful consideration and a careful judgement is needed. The random regression model estimated heritability as 0.12 to 0.20 and repeatability as 0.25 to 0.31. Except around the minimum and maximum values of age at previous calving, the heritability estimated by the random regression model was similar to that estimated by the repeatability model (Figure 2e).

### 3.3. Additive Genetic and Permanent Environmental Correlations

Estimated values of additive genetic and permanent environmental correlations tended to be lower when the difference between ages became larger. The minimum estimated value was 0.87 for additive genetic correlation and −0.16 for permanent environmental correlation (Figure 3). The estimated genetic correlation was always >0.8, the threshold proposed by Robertson (1959) as indicating a genotype-by-environment interaction [38]. In particular, the estimated genetic correlation was 0.99 between 24 and 36 months, 0.95 between 24 and 84 months, and 0.97 between 36 and 84 months of age at previous calving. These results indicate that the genetic homogeneity of the CI over age is high in Japanese Black cows. Recently, Setiaji and Oikawa (2019) estimated the genetic correlation of the CI between first and second parities to be 0.969 with the CI between second and third parities and 0.943 with the CI between third and fourth parities, and the genetic correlation of the CI between second and third parities to be 0.988 with the CI between third and fourth parities, using records of 2078 Japanese Black cows in Okinawa prefecture, the southernmost prefecture of Japan [31]. Using the random regression model, Panetto et al. (2012) reported genetic correlations between ages of cows to be <0.8 for the CI of a Guzerat dairy subpopulation in Brazil [16]. Possible reasons for the difference among the studies might be the mating pattern (artificial insemination vs. natural mating) and the selection process.

### 3.4. Selection Accuracy

Spearman’s rank correlation between EBVs obtained by both models was ≥0.97 for 36,178 cows with their own CI records and ≥0.94 for 314 sires of the 36,178 cows (Figure 4). Therefore, the results of selection based on EBVs obtained with the random regression model would be almost identical to those based on EBVs obtained by the repeatability model. With the repeatability model, the accuracy of EBV for cows increased with the number of CI records, although the gain in accuracy was smaller when more CI records were already available (Figure 5). The accuracy of EBV for sires increased with the total number of CI records of daughter cows (Figure 6). Accuracies of EBVs differed between cows with the same number of records and between sires with the same number of total records of daughter cows, mainly owing to how many related individuals had their own CI records. The minimum accuracy of EBVs for cows was similar to the theoretically expected selection accuracy. Therefore, the theoretical selection accuracy could be used as the lower limit of the accuracy of EBVs for the CI of cows with their own records obtained from the repeatability model. In this regard, we found that it would take a reasonable amount of time (several years) to obtain EBVs of cows with high accuracy for both CI and carcass traits (Figure 7). On the other hand, the age of Japanese Black bulls in Japan is usually about 5 to 6 years old when finishing on-farm progeny testing for carcass traits and selecting them as elite sires with very high selection accuracy [5,39]. Assuming that the mean of age at first calving was 24 months and the calving interval was 12 months, the CI records of daughter cows of the sires could be available when the age of the sires is about 8 to 9 years or older. Therefore, it seems to be difficult to use EBVs of the CI for sires with high accuracy at their younger age.

### 3.5. Effects of Inbreeding and Age at the Previous Calving of Cow on Calving Interval

The higher the inbreeding coefficient of cow and the age of cow at previous parity, the longer was the CI (Figure 8, Appendix A). Oyama et al. (2002) reported similar results for inbreeding coefficient and age of cow by analyzing data obtained from Japanese Black cows in Hyogo and Shimane prefectures (subpopulations), which are two western prefectures of Japan [20]. These results will be useful for determining future selection schemes and management of Japanese Black cows.

### 3.6. General Discussion

Genetically improving female reproductive efficiency is important for improved calf production and therefore efficient beef production by Japanese Black cattle. This study estimated the heritability of the CI at different ages at previous calving and the genetic correlation among ages using the random regression model. This is the first study estimating genetic parameters for the CI of Japanese black cattle in Japan using a random regression model and comparing the results with those obtained using a repeatability model. Estimated heritability was 0.12 to 0.20, consistently low and similar to that estimated by the repeatability model (Figure 2a), and the genetic correlation was always >0.8 (Figure 3). Spearman’s rank correlation between EBVs of cows with their own CI records by repeatability model analysis and random regression model analysis was almost 1. These results support the validity of the assumption that the breeding value for the CI of a cow is constant with age. We removed CI records at >120 months old at previous calving owing to the low number of records available, but the range in the age at previous calving treated here would cover a large part of the productive life of Japanese Black cows [39].

The CI records used here were based on calf shipping information. Average number of CI records per cow was 2.3, and not all cows have all their CI records, because not all progenies of a given cow are shipped to this market. In addition, the CI records of cows on farms practicing integrated management from breeding to fattening could not be analyzed. However, our results revealed information useful to managing and improving Japanese Black cow populations. For example, the heritability estimated by the repeatability model was 0.17, implying the possibility of genetically improving the CI by selection. In addition, appropriate mating decisions and management are necessary to avoiding increases in the CI due to increases in the inbreeding coefficient and the age at calving at previous parity (Figure 6). The number of records analyzed (92,019) and the number of individuals in pedigree (390,263) in this study were much higher than those in previous studies [15,16], and using the accurate, deep, and wide pedigree information in this study brought a tight genetic connectedness among individuals. These could compensate for the lower number of records per cow and contribute to a more reliable estimation of additive genetic variance (${\hat{K}}_{a}$) using the additive relationship matrix in random regression analysis in this study. Nevertheless, it is better to confirm our results by analyzing a different dataset with a higher number of records per cow.

The choice of sub-model and the assumption of heterogeneity of error variance could affect the estimation of genetic parameters by a random regression model. The Legendre polynomial is used to reduce multicollinearity and therefore to perform stable variance component estimation [40]. The results of principal component analysis for ${\hat{K}}_{a}$ supported the validity of our choice of using up to second-order Legendre polynomials as sub-models. We also performed random regression analyses using first-, third-, and fourth-order Legendre polynomials, and compared the models on the basis of the Bayesian information criterion, deviance information criterion, and total residual variance [41]. According to these three indicators, using up to second-order Legendre polynomials seemed to be more appropriate than using up to first-order Legendre polynomials. Furthermore, estimated genetic correlation was ≥0.84 when using up to first-order Legendre polynomials. On the other hand, the results about model comparison were confusing and the variance component estimation became unstable (larger SE) when the maximum order was third or fourth owing to increasing model complexity. Therefore, we reported the results obtained using up to second-order Legendre polynomials as sub-models. Furthermore, we performed analysis using a linear spline function as a sub-model, which is similar to genetic parameter estimation using a multiple trait model [42], with knots at 24, 48, and 96 months of age at previous calving, which resulted in consistently high estimated genetic correlation (≥0.97) and allowed us to draw the same conclusion. On the other hand, we also performed the analysis assuming heterogeneous error variances with the interval of 1 years (e.g., from 18 to 30 months, from 30 to 42 months, …), but the results were very similar (estimated genetic correlation ≥0.93) to those obtained assuming homogeneous error variance over age. Therefore, we reported the results assuming homogeneous error variance over age.

## 4. Conclusions

We conclude that fitting a repeatability model to multiple measurements of the CI of Japanese Black cows would make it possible to perform a valid and reasonable breeding value evaluation. On the other hand, considerable time would be required to use EBVs of cows and sires with high accuracy for selection (Figure 5, Figure 6 and Figure 7). Therefore, efficient breeding strategies should be compared, including the use of genome-wide single nucleotide polymorphism markers, as already studied for other traits in Japanese Black cattle, e.g., [43,44,45]. Furthermore, the genetic correlation of the CI with other female reproductive efficiency traits, carcass traits, and feed efficiency traits should be evaluated for determining future breeding objectives in this breed [7].

## Figures and Tables

**Figure 1 animals-11-00202-f001:**
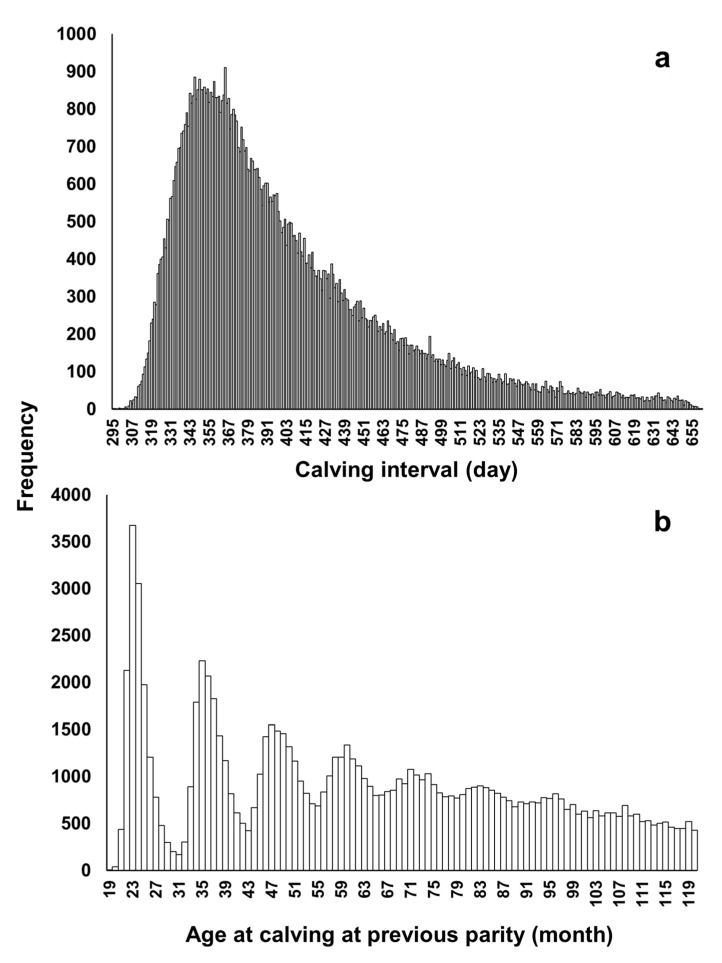
(**a**) Number of records per value of calving interval and (**b**) age at calving of cow at previous parity.

**Figure 2 animals-11-00202-f002:**
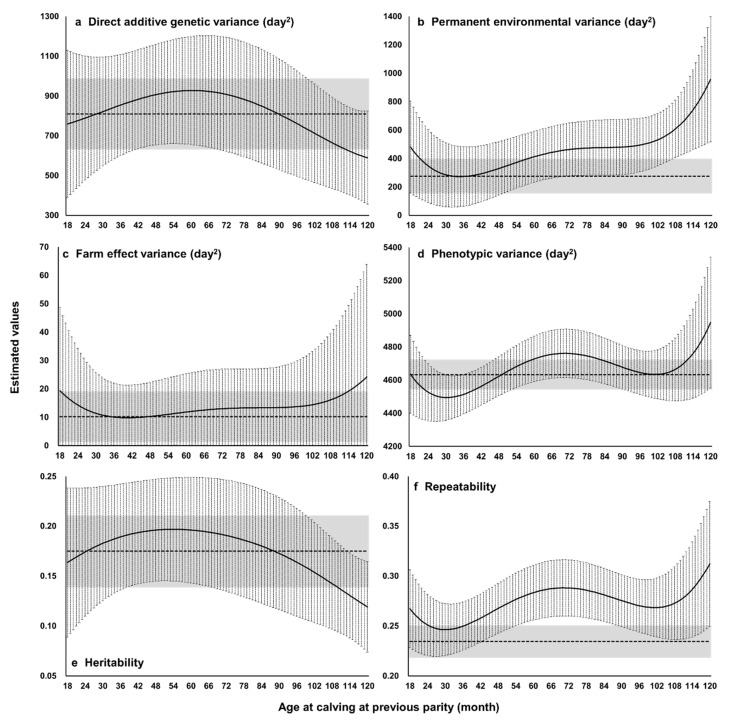
Estimated (**a**) direct additive genetic variance, (**b**) permanent environmental variance, (**c**) farm effect variance, (**d**) phenotypic variance, (**e**) heritability, and (**f**) repeatability. Curves and shading show the estimated values ± 3 standard errors: black solid with error bars, random regression model analysis; broken lines with grey area, repeatability model analysis.

**Figure 3 animals-11-00202-f003:**
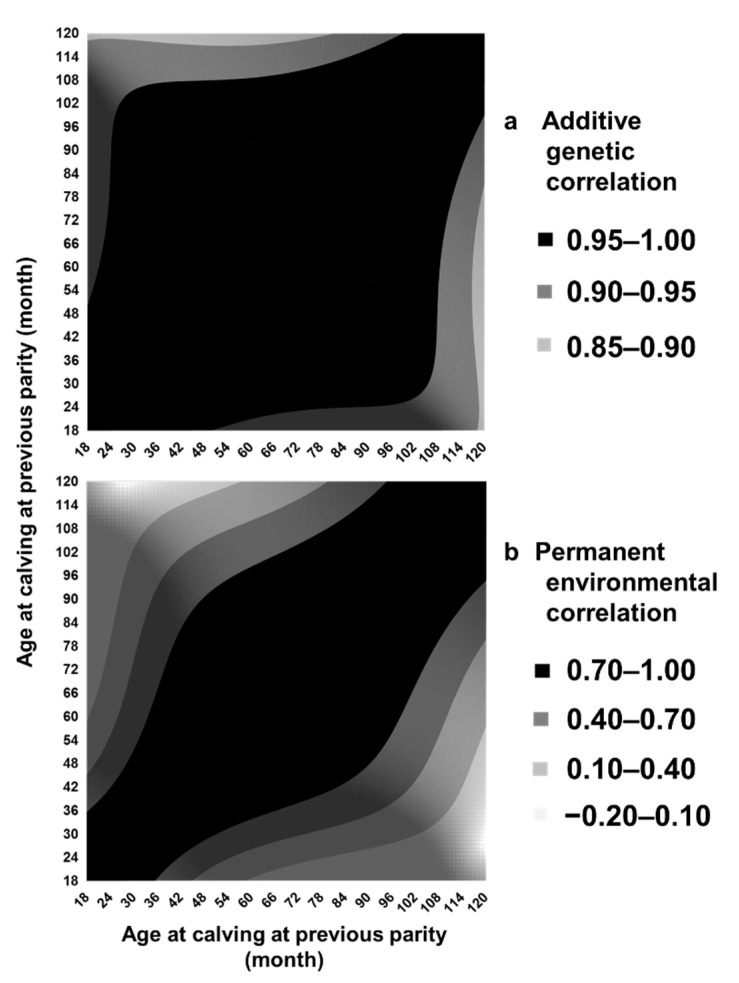
(**a**) Estimated additive genetic correlation and (**b**) permanent environmental correlation among ages of dams at previous calving by the random regression model.

**Figure 4 animals-11-00202-f004:**
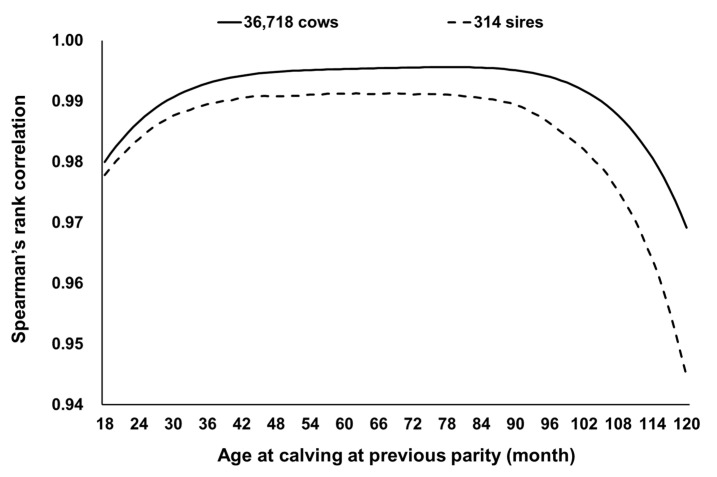
Spearman’s rank correlation between estimated breeding values obtained by the random regression model and repeatability model for 36,178 cows with their own calving interval records and 314 sires of the 36,178 cows.

**Figure 5 animals-11-00202-f005:**
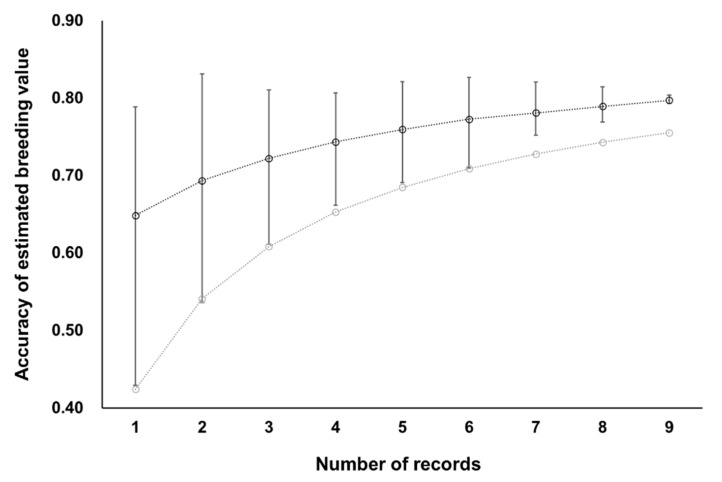
Accuracy of estimated breeding values for 36,178 cows by repeatability model analysis. Black circles and error bars show the average and its range, respectively. Gray circles show expected accuracy when the true heritability and repeatability correspond to the values estimated by repeatability model analysis in this study.

**Figure 6 animals-11-00202-f006:**
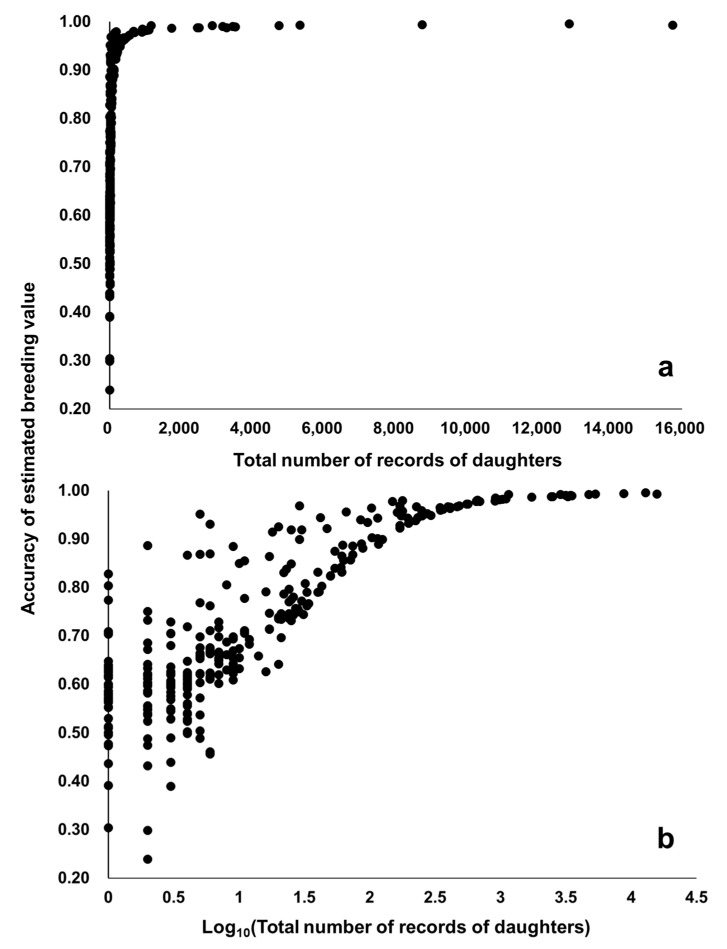
Accuracy of estimated breeding values for 314 sires by repeatability model analysis. The total number of records of daughters are in the original scale (**a**) and the logarithmic scale (**b**).

**Figure 7 animals-11-00202-f007:**
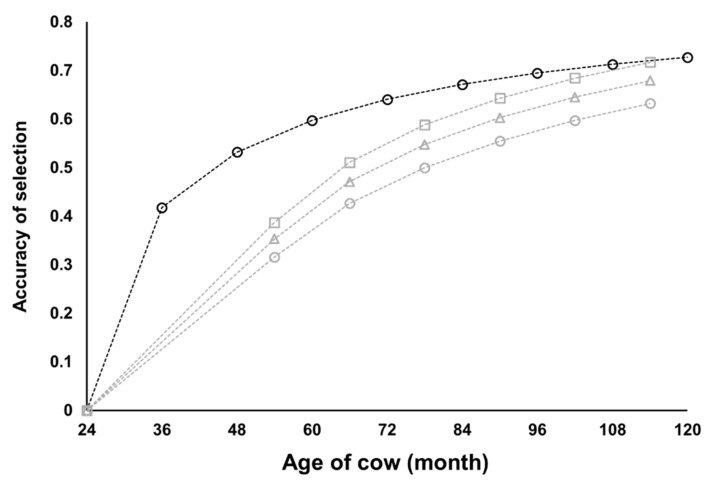
Expected accuracies of selection of cows for calving interval based on their own performance and for carcass traits based on carcass performance of their progenies. Black circles show accuracy of selection for calving interval when the true heritability and repeatability correspond to the values estimated by repeatability model analysis in this study. Gray circles, triangles, and squares show accuracy of selection for carcass traits when true heritability is 0.4, 0.5, and 0.6, respectively.

**Figure 8 animals-11-00202-f008:**
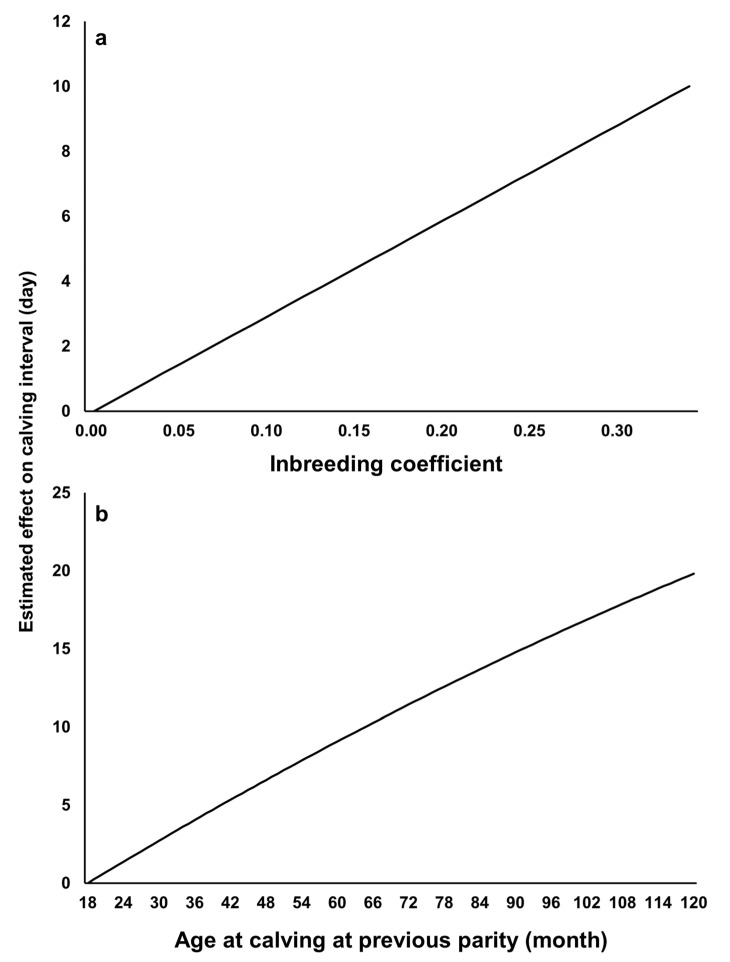
Estimated effects of (**a**) inbreeding coefficient of cow and (**b**) age at calving of cow at previous parity by repeatability model analysis.

## Data Availability

Data sharing is not applicable to this article.

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
