# Peer review of "Random Regression Analysis of Calving Interval of Japanese Black Cows"

_animals, 2021, doi:10.3390/ani11010202_

Round 1
Reviewer 1 Report
This study compared the performance of random regression model and repeatability model in genetic analysis of calving interval in Japanese Black cows. These two models are widely used in animal breeding. However, my major concern of this study is that if the number of records was enough to accurately estimate the regression coefficients for Legendre polynomials. There were averagely 2.5 records for each cow. But there were three regression coefficients to be estimated for additive genetic effect. Perhaps the data records should be filtered more strictly.
My minor comments are as follows:
- Line 102: The model in matrix notation is not very accurately expressed corresponding to the model in Line 95. The bl for Legendre polynomials in Line 98 (maybe the population mean) was not properly expressed in the matrix notation.
- Line 116: Gibbs sampling was used to estimate the variance components. A figure could be used to show if the chain was converged in Results section. Moreover, I suggest listing the estimates and SEs of variance components, at least those for additive genetic effects and permanent environmental effects, in a table.
- Line 148: The distribution of the phenotypic values was positively skewed. According to the model, the dependent variable followed normal distribution. Why not transform the data?
- Lines 225-226: It seems that the age of dam at previous parity was considered in the model for variance components estimation. Was it “year at previous calving” or “month at previous calving” (Line 97), or anything else? Moreover, why assuming the relationship between age of dam at previous calving and CI was linear?
Author Response
Thank you for your valuable comments.
Please find attached the file.

Reviewer 2 Report
The title is a little confusing because the main model is a repeatability model. Authors just used a random regression model as a validation tool. Please state clearly the objective of this study in introduction.
Authors used a random regression model to validate the repeatability model. It could be better to use a multi-trait model because a random regression model may not fit good when using few records.
Another interest is to validate a single-record model using only the first CI record. If the correlation with the repeatability is high, we don’t need to collect and use the data in later parities. It could accelerate the genetic improvement and avoid introducing selection bias as well as reducing the evaluation cost.
I wonder why authors study only on cow evaluation but not on sire evaluation. Is it because there is no sire information in 390,263 animals in pedigree? Focusing on more sire selection and reducing generation interval will increase the annual genetic gain much higher and more efficiently.
Introduction
Lines 35-36: This sentence is confusing. Wagyu cattle have 4 breeds: Japanese Black, Japanese Brown, Japanese Shorthorn, and Japanese Polled. So, what is the last one (Japanese Black cattle)?
Materials and Methods
Lines 75-76: It is not clear how the CI was calculated. Why the CI was not the days between two calvings; for example, the date at the second calving - the date at first calving. If we know two consecutive calving dates, we don’t need to know gestation length and days open, which are harder to obtain.
Lines 77-78: How many farms or what is the average farm size?
Lines 83-88: Usually, including farm-year is better than including farm and year separately in the model.
Lines 98-100: “the regression coefficient” on what?
Line 116: The gibbs3f90 program is for heterogeneous residual variances (HRV), but authors did not use HRV. Why the program was used? And why the HRV was not considered?
Results and Discussion
Lines 148-155: It will be very interesting to see the distribution of CI for each parity, maybe up to 3rd or 4th parity.
Lines 171-174: According to Figure 1 (b), the number of records at 18 and 120 months are very few, which creates artifacts (extremes) in Figure 2.
Line 186: The small farm variance (< 1%) suggests no need to treat the farm as random or no need to use random regressions unless the farm size is very small.
Lines 240-241: Yes, it is very important. However, accuracy in EBV for cows is low and takes years for the slight increase. Cow selection is very limited, compared with sire (bull) selection, which can easily reach 0.99 accuracy.
Author Response

(The authors gave the same response as above.)

Round 2
Reviewer 1 Report
Accept in present form.
Author Response
Reply to the comments by Reviewer 1
Accept in present form.
Ans: Thank you very much for your kind decision.
Reviewer 2 Report
I have a few suggestions.
There are some grammatical errors that should be corrected.
lines 296-301: When estimating heritability or additive genetic variance, we don't have to use "deep pedigree information", if authors meant "deep" as more generations or strong connections. We need more accurate pedigree information and more genetic divergence in a population.
In Figure 6, the numbers in x-axis are easier to see in the original scale (not in the log scale).
Author Response
Reply to the comments by Reviewer 2
There are some grammatical errors that should be corrected.
Ans: Thank you for your suggestion. We have checked the English grammar throughout the manuscript. The changes in the revised manuscript have been highlighted by red-colored text.
lines 296-301: When estimating heritability or additive genetic variance, we don't have to use "deep pedigree information", if authors meant "deep" as more generations or strong connections. We need more accurate pedigree information and more genetic divergence in a population.
Ans: Thank you for your comment. We have modified the expression, according to the comment by Reviewer 2 (Line 299).
In Figure 6, the numbers in x-axis are easier to see in the original scale (not in the log scale).
Ans: Thank you for your comment. We have added the figure, according to the comment by Reviewer 2.